# Near-Suicide Phenomenon: An Investigation into the Psychology of Patients with Serious Illnesses Withdrawing from Treatment

**DOI:** 10.3390/ijerph20065173

**Published:** 2023-03-15

**Authors:** Quan-Hoang Vuong, Tam-Tri Le, Ruining Jin, Quy Van Khuc, Hong-Son Nguyen, Thu-Trang Vuong, Minh-Hoang Nguyen

**Affiliations:** 1Centre for Interdisciplinary Social Research, Phenikaa University, Hanoi 100803, Vietnam; hoang.vuongquan@phenikaa-uni.edu.vn (Q.-H.V.); tri.letam@phenikaa-uni.edu.vn (T.-T.L.); 2A.I. for Social Data Lab (AISDL), Vuong & Associates, Hanoi 100000, Vietnam; 3Civil, Commercial and Economic Law School, China University of Political Science and Law, Beijing 100088, China; cu224004@cupl.edu.cn; 4Faculty of Development Economics, VNU University of Economics and Business, Vietnam National University, Hanoi 100000, Vietnam; 5Office of CPV Central Committee, 1A Hung Vuong, Hanoi 100000, Vietnam; 6Sciences Po Paris, 75007 Paris, France

**Keywords:** near-suicide, mindsponge theory, BMF analytics, patients, financial burden

## Abstract

Patients with serious illnesses or injuries may decide to quit their medical treatment if they think paying the fees will put their families into destitution. Without treatment, it is likely that fatal outcomes will soon follow. We call this phenomenon “near-suicide”. This study attempted to explore this phenomenon by examining how the seriousness of the patient’s illness or injury and the subjective evaluation of the patient’s and family’s financial situation after paying treatment fees affect the final decision on the treatment process. Bayesian Mindsponge Framework (BMF) analytics were employed to analyze a dataset of 1042 Vietnamese patients. We found that the more serious the illnesses or injuries of patients were, the more likely they were to choose to quit treatment if they perceived that paying the treatment fees heavily affected their families’ financial status. Particularly, only one in four patients with the most serious health issues who thought that continuing the treatment would push themselves and their families into destitution would decide to continue the treatment. Considering the information-filtering mechanism using subjective cost–benefit judgments, these patients likely chose the financial well-being and future of their family members over their individual suffering and inevitable death. Our study also demonstrates that mindsponge-based reasoning and BMF analytics can be effective in designing and processing health data for studying extreme psychosocial phenomena. Moreover, we suggest that policymakers implement and adjust their policies (e.g., health insurance) following scientific evidence to mitigate patients’ likelihood of making “near-suicide” decisions and improve social equality in the healthcare system.

## 1. Introduction

Diseases and injuries are natural aspects of the biological body that everyone tries to avoid. Medical treatment for illness can be very expensive and drain the financial resources of not only patients but also their families. Poor people with poor health conditions may have to face the bitter reality that they do not have enough money for treatment, thus making their diseases practically “incurable” while not technically so. They can choose either of the two bad “endings”: sacrificing their family’s finances for a chance of successful treatment or stopping the treatment and waiting for death. If a patient chooses the latter option, it is almost similar to suicide. We call this phenomenon “near-suicide”. In Vietnamese society, regarding these sad scenarios, people often say that the doctors or the hospitals “return” the patients back home so they can spend the last moments of their lives close to their loved ones. The current study is dedicated to providing the theoretical foundation and empirical validation for the “near-suicide” phenomenon.

### 1.1. The Varied Views on Suicide

Suicide is described as a death brought on by intentionally engaging in self-destructive behavior with the goal of ending one’s life [1,2]. According to the World Health Organization (WHO), every year, there around 700,000 people die from suicide. Among people aged 15 to 29, suicide ranks as the fourth most common cause of death. Nations with low and middle incomes face a greater level of threat from suicide, as the number suggests suicides in these countries account for 77% of all suicides worldwide [1]. Researchers have tried to explain suicide from multiple perspectives. Shneidman [3] considered suicide a response to overwhelming suffering, whereas Durkheim [4] stressed the significance of social interactions in suicide. Baumeister [5] considered suicide an escape from an unpleasant mental state, while Abramson et al. [6] emphasized the influence of hopelessness. More recent studies have viewed suicide through a natural–social dual lens. From a natural perspective, suicide is more likely to occur in people who have mental illnesses, physical illnesses, and substance abuse [7]. From a social perspective, some suicides are due to excessive stress, such as financial or career hardships, while others are the results of interpersonal adversities, including breakups, divorces, harassment, and bullying [8].

To define suicide, researchers have used different methods. Suicidal thoughts and behaviors were found to be influenced by functional anomalies in neurotransmitters, such as in the receptors and receptor-linked signaling networks for serotonin and norepinephrine [9]. The self-report method has also been used widely for predicting suicidal behavior [10]. When developing the fifth edition of the Diagnostic and Statistical Manual of Mental Disorders (DSM-5), experts stated five criteria for suicidal behavior disorder diagnosis: (1) Within the last 24 months, the individual has made a suicide attempt; (2) the act does not meet the criteria for non-suicidal self-injury (NSSI); (3) the diagnosis is not applied to suicidal ideation or preparatory acts; (4) the act was not initiated during a state of delirium or confusion; (5) the act was not undertaken solely for a political or religious objective [11]. This suggests that suicide is not always a mental disorder but can also be a subjectively “rational” choice as an expression of free will [12,13]. In terms of information processing, suicidal ideation (generated thoughts about the act of self-killing) may lead to suicide attempts if the perceived benefit of the self-killing act crosses a certain context-specific value threshold in one’s mind [13,14]. This aspect can be seen more clearly in cases of euthanasia [15] or suicide attacks [14].

The notion of suicide has always raised philosophical and ethical debates. John Stuart Mill argued that because the ability to make choices is a prerequisite for liberty, any action that would deprive an individual of the opportunity to make further choices should be prohibited, including suicide [16]. Jeremy Bentham suggested that although the death of a person ends their suffering, the person’s relatives and friends may grieve, and their grief may exceed said relief of suffering through suicide [17]. Thomas Szasz considered suicide a fundamental right because he argued that if freedom is self-ownership, then the right to end that life is the most fundamental of all [18]. Confucian ideology views suicide as morally acceptable and even laudable if it is committed to adhering to honor, dignity, and other virtuous values [19]. All major religions in the world generally condemn suicide with very rare exceptions because suicide is against their teachings about valuing life [14].

### 1.2. Existing Theories on Suicidality and the Data Problem in Suicide Research

Emile Durkheim categorized suicide into four different categories based on the influence of social connection and regulation: egoistic, altruistic, anomic, and fatalistic [4]. The Economic Theory of Suicide suggests that suicide is a rational economic decision based on an assessment of the financial costs and advantages of staying alive [20]. The three-step theory (3ST) proposed by Klonsky and May [21] describes the process from suicidal ideation to suicide attempts in three steps, involving the motive due to suffering and hopelessness, the protective effect of social connectedness, and the capacity for carrying our suicide attempts. The Interpersonal Theory of Suicide (ITS) [22,23] suggests that suicidal ideation is mainly caused by the coexistence of thwarted connectedness and perceived burdensomeness. The Integrated Motivational–Volitional Model (IMV) suggests that defeat and entrapment drive the emergence of suicidal ideation; the theory categorizes the process of suicide into three phases: pre-motivational, motivational, and volitional, under the influences of biosocial, motivational, and volitional factors, respectively [24]. In the Fluid Vulnerability Theory (FVT), Rudd [25] posits that suicide is dynamic and not chronologically linear. Thus, suicidal ideation may occur in episodic patterns.

While existing theories on suicidality are very helpful for psychiatrists in finding the risk factors of suicide and creating preventative measures, they often are not both highly systematic and flexible at the same time. This may cause difficulties in applications involving extreme psychological processes that are suicide-related but heavily context-specific. Moreover, most—if not all—suicide-related research has been undertaken in Western countries, and the majority of these models have been developed by authors with Westernized sociocultural backgrounds. It is not known whether suicide or suicidal behavior in other locations can be fully understood through these models. Thus, studies in non-Western countries, such as Vietnam, may enhance our understanding of suicide within understudied sociocultural environments.

In Vietnam, a series of neoliberal health policy reform measures since 1989 has affected the delivery and financing of health services, shifting from substantial government support to greater reliance on patients’ private, out-of-pocket (OOP) expenditures [26,27]. In 2010, it was reported that the average patient’s out-of-pocket hospitalization expenditures were more than USD 270 [28]. The average annual salary in Vietnam then was only USD 1684, so the hospitalization expenditure could comprise more than 16% of the annual income. The high treatment cost relative to income can lead many patients to poverty.

Studies on Vietnamese patients and the healthcare system have discovered that non-local and poor patients, in addition to those who lack access to health insurance, have an exceptionally high probability of being financially destitute (around 70%). Destitute people, together with their family members, can make collective decisions to quit treatment that very likely lead to fatal outcomes [27,29]. These decisions are not much different from the decision to die by suicide when a patient with a serious illness quits treatment due to financial burden. Without medical intervention, the chance of survival for these patients is low but not zero. Additionally, there is an unknown period of delay between the decision and the possible fatal outcome. Thus, we use the term “near-suicide” to describe this phenomenon.

The characteristics of near-suicide may be close to the notion of altruistic suicide in Durkheim’s categorization [4] but depend more heavily on the context of each family. The subjective cost–benefit evaluation is beyond individualistic economic motivation, as suggested by Hamermesh and Soss [20]. Regarding the 3ST [21] and ITS [22,23], while near-suicide is driven by the sense of burden, defeat, and hopelessness, here, the close connection between patients and their families also strongly motivates the decision. The information process of near-suicide also does not clearly follow the IMV model [24], nor is it sufficiently explained by its collectivistic rationality and interactions using the FVT [25].

Another major obstacle in psychological research on extreme phenomena such as suicide is data collection, processing, and analysis. Self-report is a widely used method. However, there is a simple fact: we cannot obtain self-reported information from someone who has completed the suicide process. In addition, obtaining self-reported information from survivors of suicide attempts is difficult and sometimes may not be a viable option in terms of ethics. Data on other suicide-related phenomena, such as suicide attacks, are limited for many reasons (e.g., low occurrence, security risks, etc.) [14]. Thus, acquiring such types of extreme data is not only difficult but also highly costly, hindering research endeavors, especially in low-income regions [30].

The phenomenon of “near-suicide” is no exception. It is not plausible to obtain information about the exact processes leading to the deaths of those who are “returned” from hospitals back to their families. We do not know specifically when, where, and how they will die. Additionally, the efforts to collect health data from these unfortunate people and their families would have a lot of ethical concerns. To tackle these issues, mindsponge-based reasoning using the information processing approach can be employed to examine the psychological mechanisms underlying certain thoughts or behaviors. Thus, this allows researchers to design surveys and utilize health data from relatively more common populations to study and predict extreme phenomena without sacrificing scientific rigor and integrity.

### 1.3. The Mindsponge Information Processing Approach for Examining “Near-Suicide”

The mindsponge mechanism was originally conceptualized by Vuong and Napier [31] to describe how a person filters information and incorporates new values into their mindset. The theory was further developed to better apply it to a wider range of processing systems and psychosocial phenomena [32]. Individuals make decisions to accept or reject information depending on the compatibility between new information and trusted values existing in the mindset. Such decisions are based on the subjective cost–benefit. The perceived costs and benefits include all related information available to the patients’ minds. The attached meanings of those pieces of information are determined by core values as well as other beliefs and knowledge stored in memory. Here, the factors considered include physical pain, fear of death, the chance of survival, the financial status of oneself and family, the future well-being of family members, the relationship between oneself and loved ones, etc. In brief, patients arrive at the decision to quit treatment if they think that the act has a positive net perceived benefit—in other words, it is deemed beneficial. This reasoning follows the mindsponge mechanism of suicidal ideation [13,14].

A decision to continue treatment is perceived as a benefit for patients because it provides a chance to prolong their existence, but there is also a financial cost for their family members. On the other hand, a decision to quit treatment is perceived as a cost for patients because it means they will likely die soon (and probably in pain due to their diseases), but their families would not lose money to pay for treatment fees. A more serious illness means a lower chance of survival without treatment but also more expensive treatment fees. In this study, we aim to examine the likelihood of patients and their families deciding whether to continue or quit treatment due to the seriousness of patients’ illnesses and their financial capacity. Thus, the research question (RQ) is as follows: How do the seriousness of the patient’s illness or injury and the subjective evaluation of the patient’s and family’s financial situation after paying treatment fees affect the final decision on the treatment process?

Regarding the raised issues about difficulties involving the health data of extreme phenomena, the research method used in this study is demonstrated as a plausible solution. We employed Bayesian Mindsponge Framework (BMF) analytics, which uses mindsponge-based reasoning for research conceptualization and the Bayesian approach for conducting analysis [33]. Together, they enabled the researchers to design and process health data for studying extreme phenomena without too much pressure due to unrealistic demands for a “perfect dataset”. More information on the technical aspects of the method is presented in Section 2 below.

## 2. Materials and Methods

### 2.1. Materials

The current study employed a dataset of 1042 patients randomly selected from various hospitals in the northern region of Vietnam [34]. These were major regional hospitals, including Viet Duc Hospital and Bach Mai Hospital in Hanoi, Haiphong’s Viet Tiep Hospital and Kien An Hospital, and Quang Ninh’s Uong Bi Hospital, to name just a few. The survey consisted of three phases. During the first phase, a total of 330 records were collected from 10 August 2014 to early February 2015. The second phase ran from February to May 2015, increasing the total number of records to 900. The final phase concluded in March 2016, with 1042 records as the final sample number. Due to the delicate nature of the research, the survey took 20 months to complete.

For the data collection, a team made up of six people was formed. One person was in charge of organizing and monitoring quality, two were in charge of entering the data into the database, and three were in charge of gathering data from the hospital sources. The interviewers contacted the patients or the patients’ family members (if the patients’ conditions did not allow them to answer) individually and gradually gathered the information for the survey, including questions regarding “sensitive data” that would be difficult to obtain in a more general/social survey. Such inquiries included the patient’s family situation, income level, extra expenses to health workers and hospital employees, and money borrowed to finance treatment. In the first phase, 1000 interviewees were randomly chosen from hospital records and based on the judgment of the data collectors regarding whether the patient/relative was available and/or willing to participate after explaining the ethical standards, issues of information nondisclosure, and the potential insights the survey may contribute to the understanding of policymakers and the general public. In some instances, the survey team had to contact patients or their families four to five times over the course of four weeks to collect a single questionnaire. As they contemplated the severity of their illnesses, some patients or their family members became too emotional to complete the questionnaire. Around 400 people took part; however, only 330 were deemed of sufficient quality. A similar procedure was also applied in the second and third phases of the survey collection.

The dataset included secondary data, which were peer-reviewed twice and passed all the ethical standards of two data journals: *Data in Brief* [35] and *Data* [34]. The dataset has been made open for reusability under the CC BY 4.0 license, so ethical approval was automatically exempted. The survey was strictly conducted in accordance with the International Committee of Medical Journal Editors (ICMJE) Recommendations, the World Medical Association (WMA) Declaration of Helsinki, and the Ministry of Health Decision 460/QD-BYT. Ethical approval was not feasible at the time of the survey design and collection since there was no authorized institute in Vietnam at the time to offer ethical approval for this type of research. The participants were informed about the ethical standards, issues of information nondisclosure, and the possible insights the survey may contribute to the understanding of policymakers and the public in general. They had to give their consent before responding to the interviews.

The recorded patients had an average age of 45.4, and the highest educational level of most of them was high school, for approximately 67.7%. The seriousness of illness or injury for most patients was reported as ‘bad’ (49.9%) or ‘emergency’ (27.4%). Most patients had a medium socio-economic status (87,14%). More than half of the patients were local residents who lived in the same regions as the hospitals (55.47%). Almost 70% of the respondents had valid insurance. Around 31.1% considered their financial conditions ‘destitute’ or ‘adversely destitute’ after paying treatment fees. One hundred nine patients had to stop in the middle of treatment (4.5%) or quit early (6.0%). Sociodemographic characteristics of patients can be seen in Appendix A.

Three variables were employed to construct the model: *End*, *Illness*, and *Burden*. The variables’ descriptions are shown in Table 1.

### 2.2. Analytical Approach

Bayesian Mindsponge Framework (BMF) analytics were selected in the current study for several reasons. BMF analytics combine the theoretical reasoning capability of mindsponge theory in dealing with hierarchical problems and the statistical power of Bayesian inference aided by Markov Chain Monte Carlo (MCMC) in fitting multilevel models [33,36]. Specifically, the mindsponge theory follows the logic of the set theory, facilitating the hierarchical organization of data [36]. Meanwhile, all forms of multilevel modelling can be deemed Bayesian in the sense that probability distributions are assigned to varying regression parameters [37], and the MCMC technique helps to overcome the complexity of hierarchical modelling and estimate posterior distributions [38].

The second reason motivating us to employ BMF analytics is its compatibility with the parsimony principle, implying that entities should not be multiplied without necessity. A parsimonious model is preferred because it facilitates the discovery of laws (or patterns of the data) and generates a more predictive conclusion on the examined issues [39,40,41]. The features of BMF analytics enable researchers to achieve parsimoniousness in constructing and fitting the model. While mindsponge theory provides a conceptual framework to aid the identification of the boundaries of studied issues, Bayesian inference allows researchers to concentrate on estimating models of the targets of interest, as it treats all properties probabilistically, including unknown parameters and uncertainties [42].

Other advantages of BMF analytics are derived from the strength of Bayesian inference. The estimates and displays of credible intervals are fundamental aspects of Bayesian analysis, which is suggested to be a better alternative for interpreting and evaluating estimated results compared to making a dichotomous decision based on the *p*-value [43]. In addition, for credible intervals, the estimated parameter is treated as a random variable, whereas the boundaries are treated as fixed. Therefore, the intervals show the likelihood that the unobserved parameter value falls inside the interval given the data at hand [44].

In order to validate the constructed model’s appropriateness and robustness, we employed a three-step validation strategy. First, we performed the Pareto smoothed-importance-sampling leave-one-out cross-validation (PSIS-LOO) to check whether the model fits the data [45]. The goodness-of-fit of the model can be divided into four levels: (1) “good” if its *k*-values are all less than 0.5; (2) “OK” if its *k*-values are greater than 0.5 but less than 0.7; (3) “bad” if its *k*-values are greater than 0.7 but less than 1; and (4) “extremely terrible” if its *k*-values are greater than 1 [46]. It should be noted that although the model’s *k*-value of less than 0.5 does not indicate that the model is the most well-specified, it helps signal that it is not under-fitted, ensuring the parsimoniousness of the model but not an oversimplification. Secondly, the effective sample size (*n_eff*) and Gelman–Rubin shrink factor (*Rhat*) were diagnosed to check the Markov chain central limit theorem. The model was deemed convergent if the *n_eff* values were greater than 1000 and the *Rhat* values were equal to 1. The model convergence was also validated graphically using trace plots, Gelman–Rubin–Brook plots, and autocorrelation plots.

The last validation step was prior-tweaking. Bayesian inference requires researchers to preset the parameters’ prior distributions before the estimation. Although we determined that presetting the parameters’ priors was uninformative to avoid subjective biases over the model-fitting process because the current research is exploratory [47], we still re-fitted the models with an informative prior distribution to test the estimated results’ robustness. If the estimated results’ using an informative prior are not much different from those of using an uninformative prior, the model’s results can be considered robust. It should be noted that uninformative priors were normal distributions with a mean value of 0 and a standard deviation of 10, specifying a flat distribution to provide the least amount of prior information as possible. The informative prior of *Burden* was preset as normal distribution with a mean value of 0 and a standard deviation of 0.5, reflecting our disbelief in the effect.

In the current study, we constructed the following model using three variables to answer the research questions stated in the Introduction.
$$End\sim \alpha \left[Illness\right]+Burden$$

The model examines whether the decision to quit early or stop in the middle of the treatment is influenced by the seriousness of the patient’s illness and the financial burden caused by treatment fee payments. The variable details are shown in Table 1.

We used the bayesvl R package to conduct Bayesian multilevel regression analysis due to its user-friendly operation method, ability to visualize eye-catching graphics, and cost-effectiveness [36]. The models were fitted with four Markov chains. Each chain included 5000 iterations, of which the first 2000 were set as warm-up iterations. All the codes and data employed in the current study are deposited in the Open Science Framework for later reproduction and further studies: https://osf.io/b9s8r/, accessed on 13 February 2023.

## 3. Results

In this section, we present the estimated results of the model constructed to examine the effects of the seriousness of a patient’s illness or injury and the financial burden after paying treatment fees on their near-suicide decision. The visualization of the PSIS-LOO test shows that all *k*-values are below 0.5, suggesting that the model is not overfitted or underfitted (see Figure 1). Thus, the estimated results can be used for later interpretation.

We examined the *n_eff* and *Rhat* values to check whether the Markov chain central limit theorem was held after fitting the model (or whether the Markov chains were convergent). As can be seen in Table 2, the *n_eff* values were larger than 1000, and the *Rhat* values were equal to 1 in both estimations using different priors, validating the convergence of the Markov chains.

The trace plots, Gelman–Rubin–Brook plots, and autocorrelation plots further confirm the convergence of the Markov chains. The trace plots in Figure 2 demonstrate “healthy” stochastic simulation processes in which the iterations were stationary and centralized around equilibriums. In Figure 3, the Gelman–Rubin shrink factors, representing the differences between the estimated between-chain and within-chain variances, drop rapidly to 1. This is a strong signal of Markov chain convergence. The autocorrelation levels of iterations can also validate the convergence. As the autocorrelation levels dropped swiftly to 0 after a certain number of lags (around 5), the memoryless property of Markov chains was held, which is another signal of convergence (see Figure 4).

From the estimated results, we found that patients facing higher financial burdens after paying the treatment fees were less likely to recover or continue the treatment ($\mu_{Burden}=-1.33$ and $\sigma_{Burden}=0.18$). In other words, they were more likely to choose a near-suicide decision. The posterior distribution of *Burden* is located entirely on the negative side of the parameter value axis (*x*-axis), suggesting that the negative association between *Burden* and *End* is highly reliable. Meanwhile, if the patient’s seriousness of illness or injury increased, they were less likely to recover or continue the treatment. Visually, the values of the intercepts’ posterior distributions decreased according to the seriousness of the illness or injury (see Figure 5), with the *Illness (Emergency)*’s posterior distribution being completely lower than the other two intercepts’ posterior distributions. This illustration confirms the reliability of the effect of the seriousness of patients’ illness or injury on the near-suicide decision. The estimated results’ patterns using informative priors reflecting our disbelief of the effect of *Burden* on *End* are not much different from those estimated using uninformative priors, implying the robustness of our estimation.

To ease the interpretation, we took the parameters’ mean values to calculate the probabilities of near-suicide decisions across patients with different levels of illness and financial burden. Mean values were used because they indicate the values at which the effect had the highest probability to happen. The following logit model was employed to calculate the patients’ probabilities for near-suicide decisions. More details about this logit model are provided by Penn State at the following website: https://online.stat.psu.edu/stat504/lesson/6 (accessed on 22 November 2022).
$$\ln\frac{\pi_{recover/continue}}{\pi_{nearsuicidedecision}}=\alpha \left[Illness\right]-1.33*Burden$$

Based on this model, for instance, we can calculate the probability of recovering or continuing treatment of patients with bad illness and suffering from destitution as follows:$$\pi_{recover/continue}=\frac{e^{6.14-1.33*4}}{1+e^{6.14-1.33*4}}=0.6942=69.42\%$$

The patients’ probabilities of recovering or continuing treatment across different scenarios of illness seriousness and financial burden are shown in Figure 6. The lower the probability of recovering or continuing treatment, the higher the probability of a near-suicide decision (the probability of stopping treatment in the middle or quitting early). The yellow line in Figure 6 illustrates the decreasing probability of recovering/continuing treatment (or increasing likelihood of near-suicide decision) when the patient or the patient’s family was becoming more financially destitute. Metaphorically, it is an “abyss of suicide”.

## 4. Discussion

In employing Bayesian analysis on a dataset of 1042 Vietnamese patients, we found that the more serious the illnesses or injuries of the patients were, the more likely they would choose to quit treatment. If they perceived that paying the treatment fees would heavily affect their families’ financial status, the effect of the illness’s seriousness on the decision to quit treatment was also intensified. Specifically, patients with the most serious health issues who thought that continuing the treatment would push themselves and their families into destitution or bankruptcy only had an estimated 24% chance of deciding to continue the treatment. On the contrary, among patients with serious health issues, those who thought that paying treatment fees would not negatively affect their financial situations had an estimated 95% chance of deciding to continue the treatment. This estimated chance was above 90% for patients with light health issues, regardless of their financial status.

Looking at the assessments of patients and their families in this life-and-death matter through the present study’s findings, we can see that they follow common patterns of rationality. This is aligned with the idea that suicide is a rational choice rather than the result of abnormal mental processes [12,13]. Rationalization is internal and subjective, which may not be similar to what is commonly expected. For example, people may assume that cold rationalization is not appropriate when it comes to the death of loved ones. However, it should also be noted that in human judgments, emotions have a very big role besides reasoning and are naturally incorporated into the process of information filtering. The final outcome of an information process of a human mind, thus, already includes influences and considerations available to the processing system at the time, be it physical, emotional, or rational [32].

Another important point to be addressed is the influence of cultural values in decision-making. Vietnamese people are deeply and simultaneously influenced by the values of different traditional schools of thought, such as Confucianism, Taoism, and Buddhism, under the phenomenon of cultural additivity [48]. These ideological values have been used as the basis for civic governance, educational examination, and social relations with a focus on family ties, fostering the collectivistic idea in Vietnam: family–village–country values, with family being the cornerstone [49]. Due to such cultural features, Vietnamese patients put much value on their families’ well-being and future in considering whether to continue treatment or not. Moreover, the current study’s data were collected in Northern Vietnam, a region deeply affected by collectivistic values, so patients’ thinking and decisions are more likely to be affected by collectivistic thinking than in other areas of the country [50]. This possibly increases their tendency to quit treatment for the whole family’s benefit. Additionally, through similar information pathways of cultural values, patients who decide to quit treatment might put their hope in “miracle cures” from traditional medicines or good luck from accumulated “virtue” [51,52]. These hidden psychosocial factors may add to the perceived benefits of quitting treatment.

The phenomenon of near-suicide has strong implications in society regarding human instincts and ethics. Patients who willingly await deaths in favor of their families’ financial well-being express altruistic intentions [4]. In addition to the common meaning of human altruism, sacrificing oneself for one’s family is also somewhat in alignment with the notion of biological altruism [53] since the near-suicide decision likely prolongs the patients’ legacies through the long-term existence of their family members.

At first glance, the near-suicide phenomenon seems to be the opposite of euthanasia. While euthanasia is health-professional-assisted suicide aiming to relieve patients from suffering due to incurable diseases [15], near-suicide is getting away from medical assistance and accepting inevitable physical suffering. However, it should be noted that in both cases, the patients chose the option that they subjectively perceived to be better than other alternatives. In the case of euthanasia, lasting pain is deemed worse than a merciful death. In the case of near-suicide, destitution for the family is deemed worse than one’s own individual demise. There are no “right” options for these poor and ill patients, only the limited freedom to choose among bad scenarios. However, such patients keep seeking the optimal choice using whatever is available at their capacity. For example, poor patients in Vietnam tend to establish co-located communities to support each other emotionally and financially [54,55]. Future studies can further explore the information-processing strategies of patients in desperate situations, in addition to sociodemographic predictors of near-suicide decisions for policy interventions.

One way to reduce the patients’ risk of facing near-suicide decisions is the lessen their financial burdens for healthcare. It is widely acknowledged that a well-functioning health insurance system can help mitigate patients’ healthcare expenditures and their poverty risks. In 2014, the Vietnamese National Assembly passed an amended Law of Health Insurance, which expands universal coverage (UC) to 100 percent of the population, covering all relevant costs in full. However, even when 60% of Vietnamese inpatients had insurance, most insured patients were not adequately covered. The effectiveness of insurance will be undermined when the insurance coverage increases, possibly leading to a greater rate of “negligible insurance” [27]. Evidently, Thuong et al. [56] discovered that heavily subsidized health insurance programs (or health insurance for the poor, near-poor, policy beneficiaries, and free HI card for some disadvantaged groups) after the implementation of amended health insurance law does not have a significant effect on reducing the out-of-pocket payments among inpatients. This finding aligns with a previous estimation that inpatients’ happiness only exists among non-resident inpatients, with a reimbursement rate of around 65% [57]. Thus, we suggest the Vietnamese government implements and adjusts its policies following scientific evidence, rather than pursuing the ambitious goal of universal coverage. Evidence-based policymaking will help the insurance system operate more effectively and allow the government to target the right disadvantaged groups (non-resident poor patients) and reduce their likelihood of making “near-suicide” decisions.

Scientific investigations first require data. Extreme psychological phenomena, such as suicide-related thoughts and behaviors, need a more dynamic method of data design and processing to be studied effectively and efficiently. The original conceptualization for the data collection leading to the dataset used in the present study was designed based on mindsponge reasoning [27]. Combined with the data processing and analysis method of the BMF when conducting new studies, this allows for the dataset’s value to be less affected by chronological changes. In other words, the data will be less likely to become “old” and, thus, irrelevant or unreliable. This is based on a fundamental property: while value systems change over time, the thinking mechanism of the human mind remains largely unchanged across hundreds of thousands of years due to its biological structures and functions. Here, it is crucial to note that although the thinking mechanism is fundamentally similar in all healthy human minds, how it turns into thoughts and behaviors is extremely dynamic and multiplex. Therefore, data design and analysis methods developed using a dynamic information processing framework, such as mindsponge, can increase the compatibility between statistical results and actual psychological processes in people’s minds [36].

Additionally, the information processing conceptual foundation allows for more accurate connections between each step in an information process, and thus, increases the reliability of predictions. These aspects are very important, particularly in cases of studying extreme psychosocial phenomena where data collection of the direct events is either impossible or highly difficult and risky [14]. Last but not least, the approach lessens the burden of the cost of science due to the race for newer data and larger sample sizes—which is even more burdensome when the data type is difficult to access and collect [30]. Therefore, with the approach demonstrated in the present study, researchers in relatively low-income situations can participate more in creating and analyzing data for studying extreme topics that used to be deemed reserved mostly for wealthy institutions.

Considering the importance of transparency in conducting and presenting scientific studies [58], we present the limitations of our study as follows. The data in this study only consisted of Vietnamese patients, so the study’s findings might be influenced mainly by the country’s sociocultural background. Further studies may investigate the near-suicide phenomenon in other countries’ populations for comparison, validation, and updating. In addition, the data were acqui36red solely in hospitals in the north of Vietnam and may not reflect features that are unique to other places. Due to geographical variances, a countrywide survey of the same kind may reveal shifting patterns. Another point worth noting is that during the COVID-19 pandemic, the perceptions and behaviors of patients and their families may be significantly influenced by the scope and magnitude of the pandemic’s impacts. Further studies on near-suicide during global health crises can provide more insights regarding large-scale changes in social situations and the healthcare system. Last but not least, due to some objective reasons, such as ethical approval exemption and no authorized institute to grant ethical approval for this kind of research in Vietnam at the time, the collected data did not receive an institutional review. However, the survey was strictly conducted in accordance with the International Committee of Medical Journal Editors (ICMJE) Recommendations, the World Medical Association (WMA) Declaration of Helsinki, and the Ministry of Health Decision 460/QD-BYT. Moreover, the data were validated and approved by *Data* (MDPI Journal) on 25 April 2019 (https://doi.org/10.3390/data4020057, accessed on 13 February 2023), while the collection protocol was validated and approved by *Data in Brief* (Elsevier journal) on 24 September 2016 (https://doi.org/10.1016/j.dib.2016.09.040, accessed on 13 February 2023).

## 5. Conclusions

We used Bayesian analysis on a dataset containing information from 1042 Vietnamese patients and discovered that patients with more severe conditions were more likely to drop out of treatment. The effect of sickness severity was amplified if patients believed that their families’ financial stability would deteriorate as a result of continuing treatment. In particular, only around a quarter of patients with the most severe health problems would choose to continue treatment if they believed that doing so would drive them and their families into poverty or bankruptcy. These findings provide supportive evidence for the near-suicide phenomenon in Vietnam, which can shed light on later studies investigating this phenomenon in other sociocultural contexts and scenarios. Moreover, based on these findings, we suggest policymakers improve the healthcare system through evidence-based policymaking to help reduce the patients’ risks of making fatal decisions.

## Figures and Tables

**Figure 1 ijerph-20-05173-f001:**
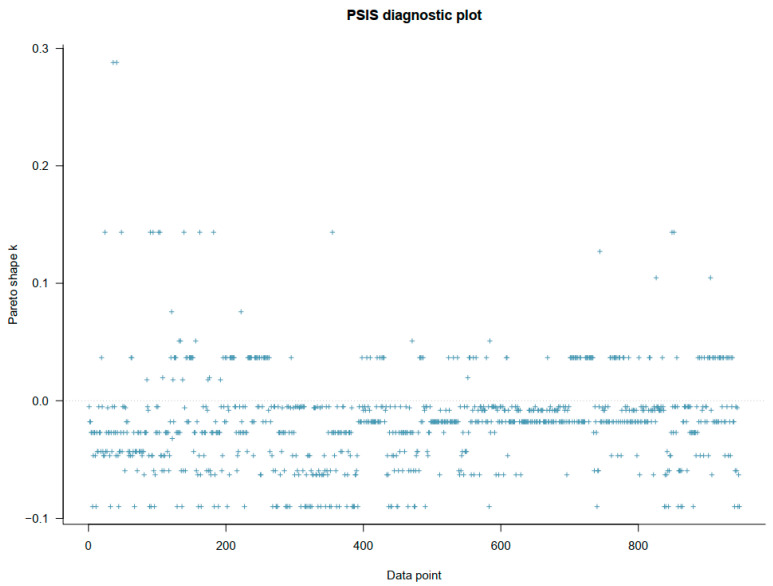
PSIS diagnostic plot of model with uninformative prior having normal distribution (0, 10).

**Figure 2 ijerph-20-05173-f002:**
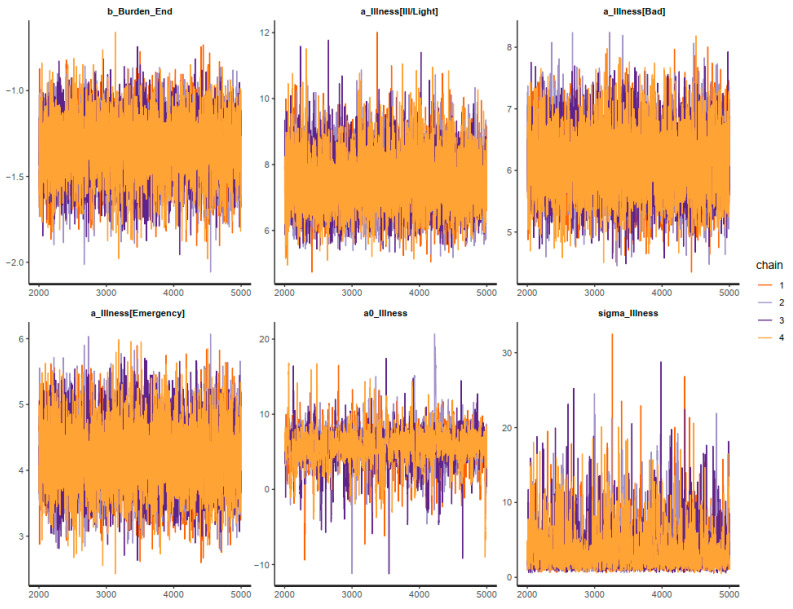
Model’s trace plots with uninformative prior having normal distribution (0, 10).

**Figure 3 ijerph-20-05173-f003:**
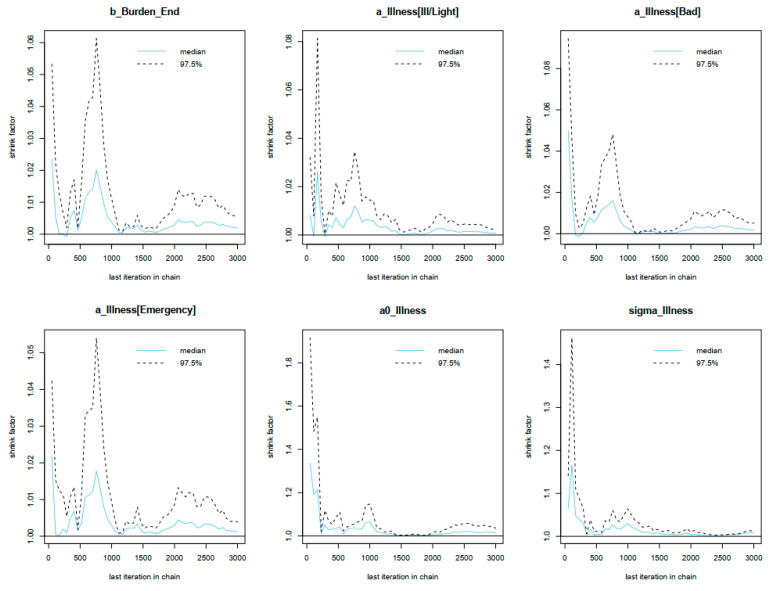
Model’s Gelman–Rubin–Brook plots with uninformative prior having normal distribution (0, 10).

**Figure 4 ijerph-20-05173-f004:**
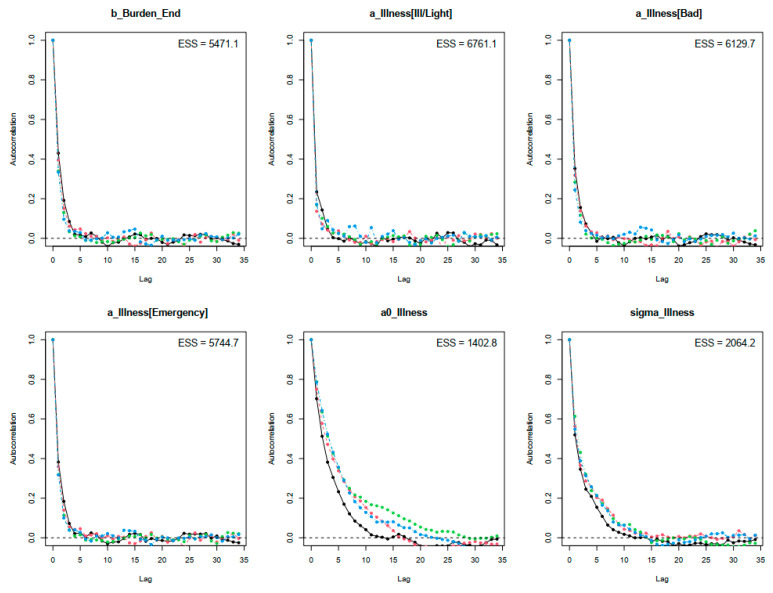
Model’s autocorrelation plots with uninformative prior having normal distribution (0, 10).

**Figure 5 ijerph-20-05173-f005:**
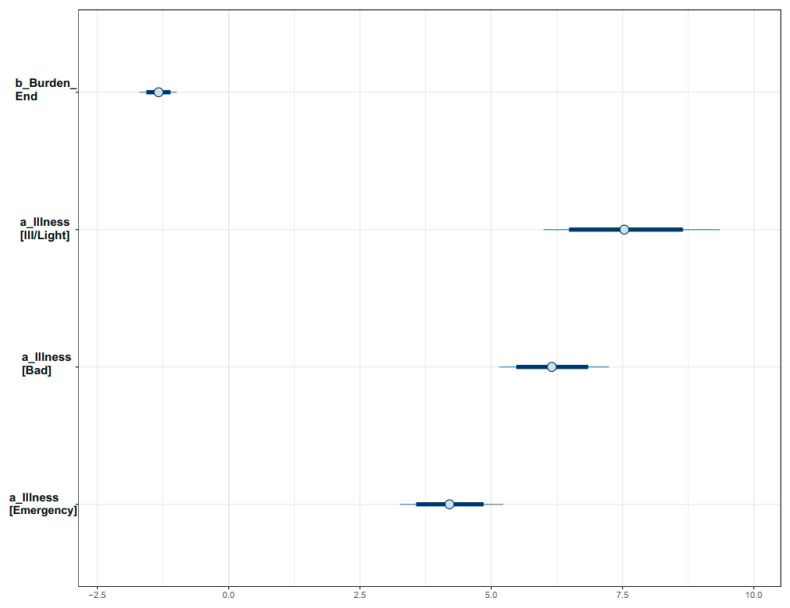
Interval plot of parameters’ probability distributions with uninformative prior having normal distribution (0, 10).

**Figure 6 ijerph-20-05173-f006:**
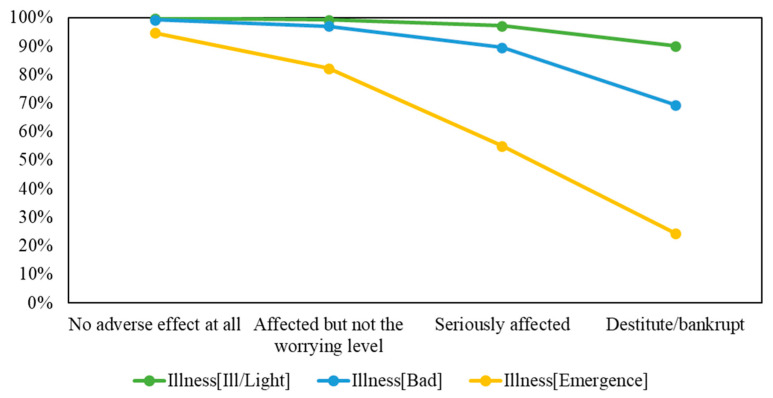
Probabilities of recovering or continuing treatment across different scenarios of illness seriousness and financial burden.

**Table 1 ijerph-20-05173-t001:** Variables’ descriptions.

Variable	Meaning	Type of Variable	Value
*End*	Outcome of the treatment	Binary	1 = ‘recovered’ or ‘need follow-up treatment’; 0 = ‘stopped in the middle’ or ‘quit early’
*Illness*	Seriousness of the patient’s illness or injury	Categorical	1 = Ill/light; 2 = bad; 3 = emergency
*Burden*	Evaluation of the patient’s and family’s financial situation after paying treatment fees	Numerical	1 = ‘no adverse effect at all’; 2 = ‘affected but not the worrying level’; 3 = ‘seriously affected’; 4 = ‘destitute/bankrupt’

**Table 2 ijerph-20-05173-t002:** Model 1′s estimated posterior results.

Parameters	Uninformative Priors				Informative Priors Reflecting Disbelief of the Effect			
	Mean	SD	*n_eff*	*Rhat*	Mean	SD	*n_eff*	*Rhat*
*Burden*	−1.33	0.18	4664	1	−1.18	0.16	4562	1
*Illness (Ill/Light)*	7.53	0.87	6211	1	7.16	0.83	6002	1
*Illness (Bad)*	6.14	0.54	5358	1	5.75	0.48	5223	1
*Illness (Emergency)*	4.19	0.51	4839	1	3.80	0.45	4975	1
*Constant*	5.64	2.39	1436	1	5.36	2.47	1192	1
*Sigma*	3.64	2.73	1952	1	3.67	2.83	1182	1

## Data Availability

The data that support this study’s findings are peer-reviewed and available on data for later replications: https://www.mdpi.com/2306-5729/4/2/57 (accessed on 13 February 2023). For convenience, all the codes and data of this study are deposited on an online repository for future validation and reproduction: https://osf.io/b9s8r/ (accessed on 13 February 2023).

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
