# Peer review of "Near-Suicide Phenomenon: An Investigation into the Psychology of Patients with Serious Illnesses Withdrawing from Treatment"

_ijerph, 2023, doi:10.3390/ijerph20065173_

Round 1

Reviewer 1 Report

This is an interesting study examining theoretical underpinnings and real-world findings on the "near-suicide" phenomenon (that is, people deciding to stop their medical treatment and assistance accepting inevitable suffering and death, in favor of their families' financial well-being) in a sample of >1000 Vietnamese subjects. A Bayesian Mindsponge Framework approach has been employed, and is well described in the manuscript. The main finding of this work is that the more serious the illnesses of patients, the more likely they will choose to quit treatment if they perceive that paying the treatment fees could significantly affect their families’ financial status.

In my opinion, this study could easily capture the reader's interest.

However, it would be better to add some considerations (especially among study limitations) in order to better contextualize and delineate study findings.

1. More information on the accessibility of the healthcare system and medical assistance in Vietnam, in order to ease the interpretation for readers that could be less familiar with differences between different countries and cultures.

2. We do not learn much on the descriptive characteristics of the study sample, so not much is told on its potential representativity. Moreover, some different variables (other than illness severity and perceived financial status) could influence people decisions (e.g. age, sex, education, household composition/number of family members, Town/rural areas, illness duration, religion, among others). If data are available, potential influence of these variables on results could be verified further. 

3. We do not know how interviews were collected and potential non-response bias (e.g. number of people contacted and number of people participating/refusing to participate), another issue which may affect representativity of study sample.

4. Since the study only includes Vietnamese participants, it could be argued that the Authors are building on results and hypothesizing a "near-suicide" construct only based on a sample with this cultural background. This could be clarified more in detail.

Author Response

Thank you very much for your constructive comments! Please see the attached file for our detailed responses.

Reviewer 2 Report

Thank you for counting on me for this review. I attach the comments in the attached document. 

It is a new and interesting topic. 

Thank you for your patience. 

Kind regards.

Author Response

(The authors gave the same response as above.)

Reviewer 3 Report

Thank you for allowing me to review this study regarding a so-called ‘near suicide’ phenomenon in Vietnam. I found it an interesting study, but I also have a few questions for clarifications, and suggestions, listed below.

Introduction

Line 43: what is the meaning of “1” at the end of the sentence?

Line 44, and other: please omit the expression “commit” suicide, as it may refer to a criminal offence and increase stigmatization of suicide. Maybe use other words, such as to die by suicide.

Please check throughout.

Authors presented an overview of theories/models of suicide. However, it can be noted that most – if not all – of these models have been formulated by authors from Westernized countries. Also, most suicide-related research has been conducted in Western countries. Thus, it is not known whether suicide or suicidal behaviour in other locations can be fully understood through these models. Thus, studies in specific countries, such as Vietnam, may contribute to enhancing our understanding of suicide within a particular socio-cultural environment. A consideration like this may enhance the rationale for the study.

Materials and methods

This section needs to provide more precise information.

Was there any reasoning behind the required sample size? Sample size calculation?

The recruitment and sampling is not very clear. If I understand it correctly, the study participants were patients who had left the hospital. Did you try to contact all patients? At what stage did you contact them? How long after they had left the hospital? How did you obtain their contact details? How many of the patients did you contact? How many refused to participate? How many had died between leaving the hospital and being contacted for the study?

If I understand it correctly, several of the patients who had left the hospital were supposed to be severely ill, and had a high risk of dying if they did not receive treatment. Yet, authors state that it took 20 months for participants to complete the survey (thus, they were still alive despite not being treated).

How did authors collect the data? Was it a survey (self-report or administered?). Interview?

Were the data provided only by the patients, or also by family members?

Please include the instrument in the paper (as an appendix?). Otherwise it is impossible to understand what data were collected.

Please include a table with sociodemographic information of participants.

Discussion

Shneidman already described that extreme feelings of being a burden to others may play a role in suicidal processes. Durkheim proposed that ‘Altruistic suicide’ was one of the four basic types of suicide. Could your findings align with these concepts?

The study seems to find that people in Vietnam place more value on the well-being of their family than on their own well-being, when deciding about continuing or discontinuing their treatment. Is this something that can be understood within Vietnamese culture? For example, in contrast, westernized countries usually place value on the individual.

The findings are very interesting, but what are the implications? Should health professionals encourage these patients to stay in hospital? Should these patients receive more home-based treatments? Or should the health insurance cover the costs of their treatment? Or just respect the patients’ decision? Other … ?

Good luck with revising the manuscript.

Author Response

(The authors gave the same response as above.)

Round 2

Reviewer 2 Report

Greetings,

The authors have made almost all the indicated changes except the format of the tables (I am attaching a template so you can see the example).

And the structure of the summary. Where they state that there are other publications where the summary is not structured. However, according to the rules of the journal and depending on the type of work, the abstract must be structured.

Regarding ethical guarantees, given the sensitivity of the data, I think they should go through an ethics committee. If not, it should be reflected as a limitation because it really is a limitation.

Congratulations again for tackling a subject of such complexity.

Best regards.

Author Response

Thanks a lot for your constructive comments! Please see our detailed responses in the attached file.
